# Entosis: From Cell Biology to Clinical Cancer Pathology

**DOI:** 10.3390/cancers12092481

**Published:** 2020-09-01

**Authors:** Izabela Mlynarczuk-Bialy, Ireneusz Dziuba, Agnieszka Sarnecka, Emilia Platos, Magdalena Kowalczyk, Katarzyna K. Pels, Grzegorz M. Wilczynski, Cezary Wojcik, Lukasz P. Bialy

**Affiliations:** 1Department of Histology and Embryology, Medical University of Warsaw, 02-004 Warszawa, Poland; imlynarczuk@wum.edu.pl; 2Department of Pathology, West Pomeranian Hospital in Gryfice, 72-300 Gryfice, Poland; mmid@wp.pl; 3HESA Association at the Department of Histology and Embryology, Medical University of Warsaw, 02-004 Warszawa, Poland; entoza1@wp.pl (A.S.); entoza2@wp.pl (E.P.); entoza3@wp.pl (M.K.); 4Nencki Institute of Experimental Biology, Polish Academy of Sciences, 02-093 Warsaw, Poland; k.pels@nencki.edu.pl (K.K.P.); g.wilczynski@nencki.gov.pl (G.M.W.); 5US Cardiovascular, Amgen Inc., One Amgen Center Drive, Thousand Oaks, CA 91320-1799, USA; cwojcik@amgen.com

**Keywords:** entosis, cancer, cell internalisation, cell-in-cell, cell adhesion, cancer prognosis, cancer predictor factor

## Abstract

**Simple Summary:**

We review published clinico-histopathological studies establishing entosis an important prognostic and predictor factor in various cancer types. We also propose a new model to study this phenomenon, which involves active entry of one cell into another one. The internalized cell can remain viable and leave the host cell after a long time, potentially leading to cancer recurrence. Entotic figures are cell in cell structures, in which the nucleus of external cell is crescent-shaped, and the inner cell is surrounded by the extensive space entotic vacuole, distinguishing entosis from cell cannibalism. Entosis correlates with cancer worse prognosis in head and neck squamous cell carcinoma, anal carcinoma, lung adenocarcinoma, pancreatic ductal carcinoma, and some breast ductal carcinoma. The BxPC-3 pancreatic cancer cells provide a new, more convenient model for entosis research in comparison to the previously described semidherent MCF7 model. BxPC-3 cells undergo and survive spontaneous entosis in normal adherent culture conditions.

**Abstract:**

Entosis is a phenomenon, in which one cell enters a second one. New clinico-histopathological studies of entosis prompted us to summarize its significance in cancer. It appears that entosis might be a novel, independent prognostic predictor factor in cancer histopathology. We briefly discuss the biological basis of entosis, followed by a summary of published clinico-histopathological studies on entosis significance in cancer prognosis. The correlation of entosis with cancer prognosis in head and neck squamous cell carcinoma, anal carcinoma, lung adenocarcinoma, pancreatic ductal carcinoma and breast ductal carcinoma, is shown. Numerous entotic figures are associated with a more malignant cancer phenotype and poor prognosis in many cancers. We also showed that some anticancer drugs could induce entosis in cell culture, even as an escape mechanism. Thus, entosis is likely beneficial for survival of malignant cells, i.e., an entotic cell can hide from unfavourable factors in another cell and subsequently leave the host cell remaining intact, leading to failure in therapy or cancer recurrence. Finally, we highlight the potential relationship of cell adhesion with entosis in vitro, based on the model of the BxPc3 cells cultured in full adhesive conditions, comparing them to a commonly used MCF7 semiadhesive model of entosis.

## 1. Introduction

Entosis refers to the invasion of one living cell into another of the same type with involvement of adhesion molecules, actin cytoskeleton and expenditure of energy [1,2]. The term entosis was first defined by Overholtzer in 2007, as a new type of cell death [2]. Entosis is derived from the Greek word that means “inside” or “within”, based on the observation that one viable cell invades the other viable cell. To simplify the nomenclature in this article, we refer to the internalized or engulfed cell as “inner”, while the internalizing or engulfing cell is referred to as the “outer” cell. Entosis is characteristic for epithelial cells and epithelial cancers, and is triggered by detachment of cells from the basement membrane. As a result of entosis, characteristic cell-in-cell (CIC) structures are formed [3]. Once engulfed, entotic cells can be eliminated through regulated cell death within the entotic vacuole (entosome), via a specific autophagy-related process, commonly known as LC3-associated phagocytosis (LAP) [4]. This process is independent of the apoptotic pathway [2]. However, various fates of entosis are possible (see Figure 1). The inner entotic cell can undergo division within the host cell or can escape from the host cell, without any signs of degeneration [1,2]. Therefore, the European Cell Death Organization (ECDO) questioned whether entosis is a form of cell death, since one cell can exist for an extended time within the other one [5].

Entosis is neither a type of phagocytosis nor cell cannibalism. During entosis, the inner entotic cell actively enters into the host cell through activation of Rho proteins, followed by formation of adhesion bonds and actomyosin filaments [6]. After cell invasion into the host cell, the inner cell is surrounded by a double membrane of the entotic vacuole, with an extensive space between membranes. The presence of the entotic vacuole is a feature distinguishing entosis from cell cannibalism, in which the phagocytosed cell is surrounded by a thin double-membrane space, with subsequent digestion of the inner membrane [2]. The fate of the two cells undergoing entosis can involve death of the inner cell, death of the outer cell, death of both cells, and survival of both cells. New clinico-histopathological studies prompted us to summarize the significance of entosis in cancer, as entotic figures can nowadays be considered a new predictor in routine histopathological evaluation [7].

Therefore, in this review we summarised published clinico-histopathological studies on the significance of entosis and its relationship to cancer prognosis. We also showed that some anticancer drugs could induce entosis in cell culture, either as a cell death or as a pro-survival escape mechanism. At the end, we highlight the potential relationship of cell adhesion with the fate of entotic cells in vitro, based on the model of BxPc3 cells.

## 2. General Mechanism of Entosis

The integrity of all epithelial tissues depends on intact adherens junctions. They are formed by an assembly of cadherins spanning the intracellular space and cytoplasmic catenins anchoring their cytosolic tails that are coupled with actin filaments, and further depend on actin–myosin contraction that is regulated by Rho GTPases [8]. The known molecules involved in entosis are summarized in Figure 2. During entosis, Rho activation occurs in a cell that has lost contact with the basement membrane. Subsequently, the cell produces adherens junctions with the neighbouring cell and actively penetrates it, creating a CIC structure [2]. Upon initiation of entosis, E-cadherin and β-catenin accumulate at the surface of the internalizing cells. Further, actomyosin accumulates at the cell cortex. This process involves the Ras homolog family member A (RHOA) as well as kinases ROCK1, ROCK2, (Rho associated coiled-coil containing protein kinase), and DIAPH1 (diaphanous related formin 1) [2,9]. Actomyosin complexes, together with regulatory molecules, provide the motor mechanism that promotes cortical plasma membrane budding that is responsible for invasion into the host cell. This invasion is driven by actin and undergoes regulation by myocardin-related transcription factor (MRTF), serum response factor (SRF), and ezrin [2,10]. The regulation of microtubules by Aurora kinase was also shown in the context of entosis [11]. A distinctive mechanism was demonstrated in cells undergoing glucose withdrawal. Cells that had a lower AMPK (5′ AMP-activated protein kinase) activity did not undergo entosis, as opposed to those cells with a higher AMPK activity [12]. In this context, cells that were less metabolically efficient were digested within the more metabolically efficient ones. In this way, recovery of metabolic components was possible by sacrificing cells that were less likely to survive.

### Entosis, Mitosis, Aneuploidy and Cell Competition

Mitosis can drive cancer cell internalisation through entosis and is linked to the cell division control protein 42 homolog (CDC42), a small dimeric GTP-ase of the Rho family, which is overexpressed in several cancer types [12]. This specific form of entosis that occurs during mitosis is called entotic mitosis. It can be found in cells from different cancer types. This kind of entosis is initiated by mitotic de-adhesion of cells and is associated with exposure to antimitotic agents or CDC42 depletion. CDC42, a regulator of epithelial cell biology, can control the CIC formation. It was shown that CDC42 depletion in epithelial cells in suspension did not affect entosis. However, its depletion in adherent cells caused the formation of CIC structures. Moreover, RhoA activity could affect aberrant mitosis in cancer cells [12]. Entosis also causes aneuploidy of the host cell, making it more malignant. This results from the mechanical interference of the mitotic spindle with the entotic vacuole and inadequate cytokinesis of the outer cells [13,14].

It is postulated that entosis is a mechanism of cell competition. During entosis, the host cells can obtain nutrients from inner entotic cells [9,15]. In this situation, the internalizing outer cell is referred to as the “winner” and the internalized inner cell is called the “loser”. This kind of competition mechanically differentiates cells into stiffer cells to be digested by softer ones that are present in their vicinity. Stiffer cells by activation of Rho and actomyosin filaments can invade into the softer ones [12,16].

## 3. Entosis in Cancer

Entotic cell death is postulated to act as a tumour suppressor mechanism by facilitating the death of entotic cancer cells [8,9,17,18,19]. However, entosis can lead to aneuploidy and polyploidy, which promote tumour progression. Moreover, multinucleation is a common feature observed in internalizing host cells, following matrix deadhesion [14], mitosis [12], and glucose starvation [12]. Furthermore, entosis does not always lead to the death of entotic cell. Thus, in certain situations, entotic inner cells remain viable, proliferate within the outer host cell, and can eventually escape from the outer cell. However, the reasons determining the fate of each inner entotic cell remain unknown.

An increased number of entotic structures correlates with tumour promotion and progression [20]. Thus, environmental conditions like lack of glucose are linked to an increase in the grade of genetic alteration. Interestingly, entosis can also indicate tumour suppression with the elimination of aberrant constituents [12]. Activated KRAS, one of the most prominent oncogenes, can stimulate entosis [9]. It is known that expression of cadherin proteins, E-cadherin or P-cadherin, induces entosis in human cancers [9].

Cancer tissues with numerous entotic figures represent a more malignant phenotype than the same variant without such structures [21,22]. Thus, entosis is likely beneficial for malignant cells. For instance, an internalizing cell can hide into the other, from environmental factors like chemotherapy, antibodies, cytotoxic T cells or malnutrition. Indeed, some anticancer drugs and substances induce entosis [12,23,24]. Moreover, in the case of prostate cancer cells, entosis was shown to be an escape mechanism from the anti-cancerous effects of the tyrosine kinase inhibitor nintedanib [24].

There was only one case in which CIC structures were associated with reduced metastasis in pancreatic cancer, which is the opposite to other tumour types [25,26]. However, one should keep in mind that the fate of entosis can be affected by the interactions of cancer cell and tumour extracellular matrix [27]. As the majority of cancers entosis correlates with poor outcome recurrence of disease [1,27], it can also be hypothesized that the inner entotic cell can survive unfavourable conditions caused by anticancer drugs within the host outer cell, protected by the environment of the entotic vacuole. Subsequently, the internal cell can leave the outer cell after a long time [2]. Thus, both cells can survive entosis, leading to a failure in the therapy and recurrence of cancer.

### 3.1. Clinical Relevance of Entosis in Cancer

Entosis is now regarded as a potential new predictor factor in cancer prognosis, as shown in the clinico-histopathological studies summarised in Table 1.

One of the biggest clinico-histopathological studies involving 516 patients with pancreatic ductal cancer (PADC) that was recently published [7] showed a direct correlation between the number of entotic CICs found in histopathological specimens with clinical outcomes. It was demonstrated that an increased number of entotic figures was independently associated with poor prognosis. Moreover, entotic figures were more prevalent in poorly differentiated PADC and in metastases (especially in the liver).

In another recent study it was also shown that entotic figures represent 91% of the CIC structures in samples of breast ductal carcinoma collected from untreated patients, at baseline [29]. In this clinico-histopathological study, entosis was shown to be an independent prognostic factor. However, the prognosis differed, depending on the subtype of CICs and subtype of cancer (i.e., low number of CICs correlates with shorter survival of Luminal B (Her2+) patients, but no other investigated cancer subtypes. Two earlier clinico-histopathological studies showed that in squamous cell carcinomas of the head and neck, presence and intensity of entosis is a better predictor of clinical outcome than apoptosis or senescence [21,22]. One study revealed the poor outcome and disease recurrence of lung adenocarcinomas, associated with an increased number of entotic structures [28]. In the case report of one patient with invasive ductal carcinoma, a high frequency of entotic figures was accompanied by poor prognosis [30].

An interesting link between loss of cellular adhesion to extracellular matrix, cancer progression and entosis was found in the case of epithelial ovarian cancer (EOC) [31]. An analysis of 268 archived paraffin-embedded samples found that loss of the 4.1N protein was associated with more malignant phenotype and worse prognosis. The 4.1N protein linked plasma membrane to cytoskeleton, thus loss of 4.1N disrupted integrin-dependent cell adhesion and associated signalling. Entosis in cancer cells cultured in vitro, induced by the loss of 4.1N, was associated with cancer cell resistance to cell death. This contrasted with the typical response of non-malignant cells, which often undergo a specialised type of cell death (anoikis) after losing adhesion to the extracellular matrix. Even though the authors did not calculate entotic figures in specimens, it could be concluded that loss of 4.1N protein induced entosis in vitro and was associated with worse prognosis in vivo.

The association of entosis with cancer progression could be explained by the hypothesis that it has a dual role—while entosis can lead to cell death, it can also promote cell survival [32]. Moreover, it was recently shown in vitro that both inner and outer entotic cells could synthesize DNA (BrdU incorporation) and enter mitosis during entosis [33]. At present, it is not clear what factor(s) determine whether entosis will lead to cell survival or cell death. Elucidation of this pathway might open new avenues for cancer therapy in the future.

It is known that an anticancer drug, paclitaxel, induces entosis in breast cancer cell line MCF7 [12]. Moreover, methylselenoesters induce death of inner entotic cells of Panc-1 pancreatic cancer cells, through downregulation of CDC42 and β1-integrin (CD29) [23]. Recently, it was shown that prostate cancer cells develop entosis as a pro-survival mechanism, upon treatment with the tyrosine kinase inhibitor nintedanib [24]. All these findings open new perspectives for the investigation of entosis in molecular medicine research, as a target for novel anticancer strategies.

### 3.2. Entosis Models for Cancer Research

Originally, entosis was described in the MCF10A human mammary epithelial cells and MCF7 human breast cancer cells cultured in non-adherent conditions [2]. However, many cell lines were used in research on entosis in cancer—prostate cancers, pancreatic cancers, and breast cancers [11,23,34,35,36]. Therefore, there is a need to develop a uniform model for entosis detection. Easy and reproducible visualisation of entotic figures in cell culture was achieved by a simple haematoxylin stain, for observation with a light microscope, with additional advantage that the slides could be stored for a long time (see Figure 3; Appendix A). Counting the number of entotic cells and describing the morphology of CIC structures could be verified by an independent pathologist even after a long time, in archival slides of a specimen collection.

Entosis taking place in a tumour in vivo varies from what is observed in cancer cell lines in vitro. In cancer cell lines, entosis was described almost only in cells, either artificially detached from a substratum by culturing them in nonadhesive conditions (MCF7 cells), after starvation [12], or due to the effects of anticancer drugs such as paclitaxel [12], nintedanib [24], and novel methylselenoesters [23]. Only non-cancerous HaCaT keratinocytes were shown to undergo entosis as adherent cells, but the rate of entosis was very low (less than 0.05%) [37]. In contrast, in in vivo samples of tumours, entosis can be found under normal circumstances, without any anticancer treatment, both in primary lesions and in metastases (see Figure 4).

Therefore, there is a need to find more physiological models for spontaneous entosis in cancer for in vitro research, which would be more adequate to the situation observed in tumours in vivo. Among various cell lines cultured by our team, we identified pancreatic cancer cells BxPC3 as competent for spontaneous entosis in normal adhesive culture conditions [38]; (see Figure 3 and Figure 5; Appendix A).

As mentioned earlier, the first cancer cells shown to be entosis-competent are MCF7 human breast cancer cells cultured in non-adherent conditions [2]. In our laboratory, we observed entosis in BxPc3 pancreatic cancer cells, which strongly adhere to the bottom of the culture flask [38]. However, among these cells we observed about 3% of entotic CIC structures of diverse morphology (Figure 3 and Figure 5). The majority of the observed entotic cells demonstrated no signs of degeneration, such as fragmentation of chromatin, karyolysis, or cell membrane rupture. Overholtzer originally defined entosis as a type of cell death [2]. In his model, cells were cultured in suspension or in the presence of EDTA, to ensure detachment from substrate. We also tried to repeat conditions provided by Overholtzer for the MCF7 cells, using BxPC3 pancreatic cancer cell line. Despite our attempts, we failed to obtain BxPC3 cells loosely connected to the growth surface, which would remain rounded during the incubation time. In plastic vessels designed for suspension cells, BxPC3 cells adhered strongly and were flattened within 2 h of seeding. We were, therefore, unable to obtain rounded semi-adherent BxPC3 cells. When incubated with EDTA (starting from 1 µM concentration) BxPc3 cells died separately through anoikis within 12 h, and therefore, failed to execute entosis. Thus, the conditions used by Overholtzer for the MCF7 cells using the pancreatic cancer cell line BxPC3 cells did not allow us to obtain a detached viable BxPc3 in suspension.

Detailed studies on CIC structures on adherent BxPc3 cells were performed through a confocal tile scan of specimens (Figure 5; Appendix A) or classical light microscopy (Figure 3; Appendix A). Cells from 30 fields were counted and analysed by two independent researchers, for the total number of CIC structures, mitotic figures, and regressive changes (i.e., degeneration, both nucleus, or cytoplasm alternations). Detailed morphology of the entotic cells is shown in Figure 5.

Entosis in BxPc3 cells differs from the MCF7 model (Table 2). The adherent BxPc3 cells demonstrated overall ~10× less entotic figures (Figure 3) than the MCF7 cells in artificial non-adherent conditions (3% vs. 30%) [1,2,38]. Moreover, in the MCF7 cells, ~70% of inner entotic cells died within the entotic vacuole, due to activation of the lysosomal pathway [1,2]. In contrast, in the BxPc3 cells, ~80% of inner entotic cells demonstrated no features of regressive changes. (Figure 3 and Figure 5A,B). Interestingly, the observed ratio of inner entotic cells entering mitosis (Figure 5C) was similar in both cell models.

## 4. Conclusions

Entosis is frequently found during histopathological examination of specimens obtained from various human cancers. Entotic cell death was postulated to act as a tumour suppressor, through elimination of aberrant cells. Indeed, some anticancer drugs induce entosis. However, the increased number of entotic structures correlates with tumour promotion and progression. Cancer tissues with numerous entotic figures usually represent a more malignant phenotype, which could result from the beneficial effects of entosis for malignant cells. For instance, internalised cells could be protected by the entotic vacuole formed inside the host outer cell, from harmful environmental factors like chemotherapy or other unfavourable conditions induced by anticancer drugs. Subsequently the inner entotic cell can leave the outer cell intact, after a long time. Such process can potentially result in chemotherapy failure or even cancer recurrence. Indeed, it was shown that, prostate cancer cells appear to develop entosis to survive treatment with the tyrosine kinase inhibitor nintedanib. Poor outcome and cancer recurrence associated with an increased number of entotic figures was proven in clinico-histopathological studies of head and neck in squamous cell carcinomas, anal carcinoma, lung adenocarcinomas, and pancreatic ductal carcinoma. All these findings open new perspectives to investigate entosis as a potential target for novel anticancer strategies.

We compared a fully adherent in vitro entosis model of BxPc3 cells with a semiadhesive model of MCF7. We noted both similarities and differences to entosis induced in MCF7. Importantly, in the BxPc3 model, most inner cells demonstrated no signs of degeneration or cell death. We postulated that this model represents spontaneous or physiological entosis.

Due to the dual role of entosis in tumour progression, there is an unmet need to introduce a clear and reproducible criteria to identify and describe entotic cells to unify research of this phenomenon. In this context, we propose to unify documentation of entosis with an easy and reproducible method, using the standard haematoxylin stain for light microscopy. This allows counting the number of entotic figures as well as evaluation of the morphology of cell structures. It is a simple and inexpensive method, creating permanent specimens that could be viewed and evaluated repeatedly by different laboratories, as well as be easily used for investigation of different agents affecting entosis.

## Figures and Tables

**Figure 1 cancers-12-02481-f001:**
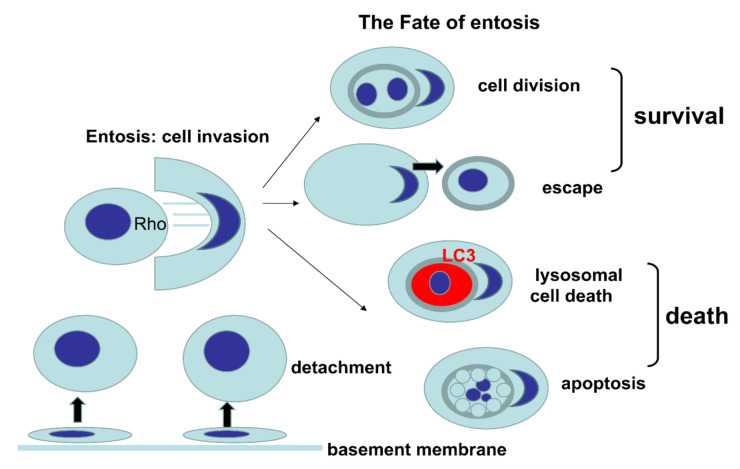
Graphical presentation of various fates of entosis.

**Figure 2 cancers-12-02481-f002:**
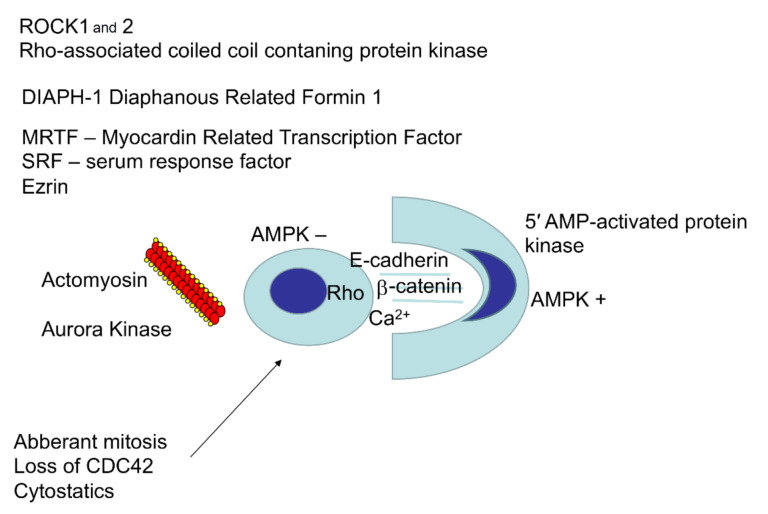
The main known molecules regulating entosis. The central part represents a schematic view of the cell–cell interaction during entosis. On the left, the interior cell is rounded after losing its contact with the substrate. On the right, the outer cell wraps around the inner cell and has a crescent-shaped nucleus. Both cells interact through formation of adhesion junctions mediated by E-cadherin, β-catenin, and calcium ions. The protein kinase AMPK is active in the outer cell and inactive in the inner one. On the left, there is a list of molecules that are active in the inner cell. The arrow indicates factors inducing entosis in cancer cells.

**Figure 3 cancers-12-02481-f003:**
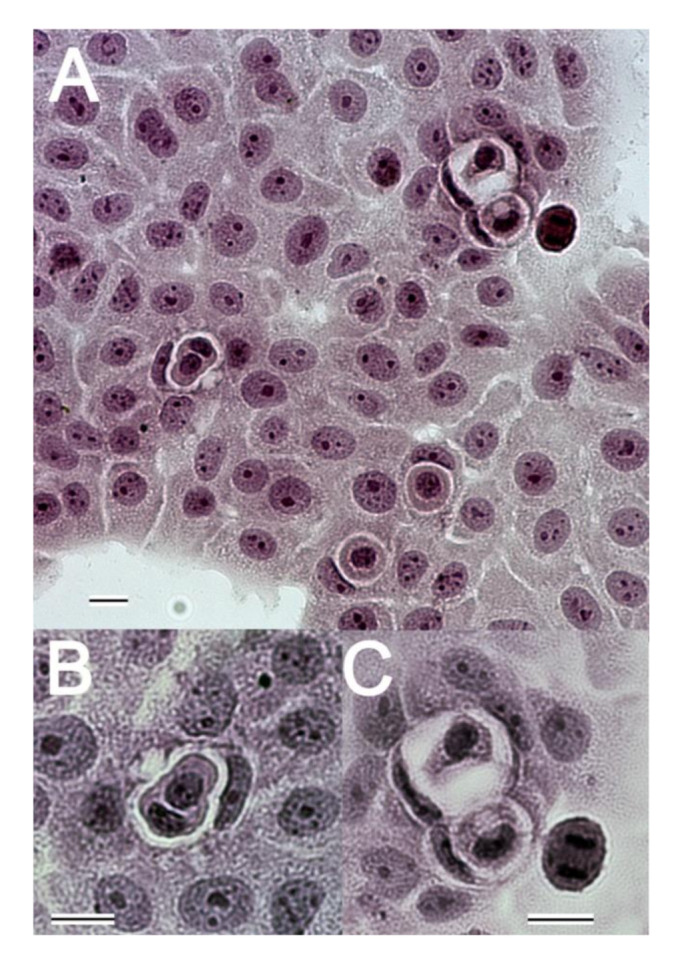
Photograph of pancreatic cancer cells BxPC3 cultured on coverslips stained with Haematoxylin. (**A**) General view of BxPc3 cells with entotic structures. For about 100 cells visible in the field of view, five demonstrate three characteristic features of entosis: (a) CIC structure (one cell is within the second one); (b) the outer cell is stretched with a crescent-shaped nucleus; and (c) the characteristic entotic vacuole is formed around the inner cell. The outer cell demonstrates normal nuclei with sharp edges and well-visible nucleoli, without features of chromatin condensation or chromatin vacuolisation. The cytoplasm of the host cells is stretched and seems to be involved in formation of the big vacuole that surrounds the inner cell. The internal cell has preserved the cytoplasm, its nucleus is well delimited without regressive changes; (original magnification 100×) (**B**,**C**) Selected entotic structures. (**B**) Double cell-in-cell structure; (original magnification 400×). (**C**) Two entotic figures with huge entotic vacuoles; (original magnification 400×). All scale bars: 10 µm. Data by Izabela Mlynarczuk-Bialy (M.D., Ph.D.) Magdalena Kowalczyk (M.Sc.).

**Figure 4 cancers-12-02481-f004:**
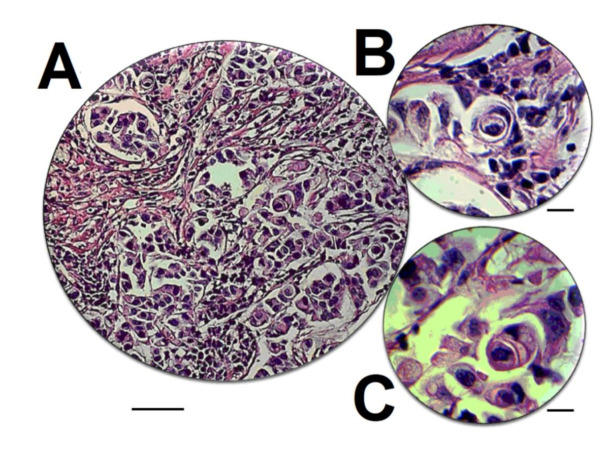
Entosis detected in metastases of double-negative ductal breast carcinoma into axillar lymph node. (**A**) Subcapsular region of axillar lymph node; original magnification 200×. (**B**,**C**) Selected entotic figures form picture (**A**); original optical magnification 200× and digital magnification 2.5×. Scale bars: (**A**) 50 µm; (**B**,**C**) 10 µm; (H&E stain). Data by I. Dziuba (M.D., Ph.D.).

**Figure 5 cancers-12-02481-f005:**
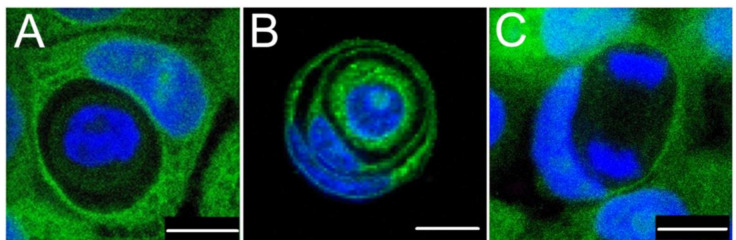
Entosis in BxPC3 cells. Entosis without degeneration (**A**). One single cell is surrounded by the host cell. Membranes of both cells remain intact; the inner entotic cell seems to be localized within a large vacuole formed by the host cell (**B**). Double Cell-in-Cell structure. The internalizing cells are stretched and have a narrow area of the cytoplasm. Their nuclei have a crescent shape (**C**). Metaphase of mitosis of an inner entotic cell inside a cell that exhibits entosis-like appearance, such as the crescent nucleus and stretched cell membranes with a narrow region of cytoplasm. Confocal scan by SP5 Leica confocal microscope. Cytoplasm is stained with CFDA-SE in green and nuclei with DAPI in blue. Scale bar: 10 µm. Data by Izabela Mlynarczuk-Bialy (M.D., Ph.D.), and Agnieszka Sarnecka (M.D.), and Lukasz Bialy (M.D., Ph.D.).

**Table 1 cancers-12-02481-t001:** Clinico-histopathological studies of entosis in human cancers.

Type of Cancer	Number of Cases	Conclusion	Year	PI	Ref.
Head & neck anal rectal squamous cell carcinoma (SCC)	416 (total cases)	Entosis is an independent prognostic factor in all studied cancers	2015	Schwegler	[22]
	321 (head & neck) 23 (anal)	Entosis is associated with worse prognosis of head & neck and anal carcinomas			
	82 rectal	Entosis is associated with better prognosis in rectal carcinomas			
Head & neck SCC	201	Entosis is a predictive marker associated with worse prognosis	2017	Schenker	[21]
Lung adenocarcinoma	273	Entosis is associated with earlier recurrence and patients’ death	2018	Mackay	[28]
Breast ductal carcinoma	160	Entosis is an independent prognostic factor. However, the prognosis differs depending on the cancer subtype	2019	Zhang	[29]
Pancreatic ductal cancer	516	Entosis is an independent prognostic factor associated with worse prognosis	2020	Hayashi	[7]

**Table 2 cancers-12-02481-t002:** Differences and similarities in entosis between MCF7 and BxPc3 cells.

Parameters	Forced Entosis, MCF7-Cells, Semi-Adherent Conditions [1,2]	Spontaneous Entosis, BxPc3 Cells, Strong Adhesion [38]
	Differences	
EDTA or EGTA	Cells are alive and remain rounded	Cells die in the presence of EDTA starting from 1 µM
Suspension culture flasks	Alive cells remain rounded	Alive cells adhere to the bottom
Fate of entosis	70% death of entotic cells	80% CICs without degeneration
	Similarities	
Cell division of inner cell	Observed	
Duration of entosis	Start 6 h max number of entotic features 12–24 h, decrease in entosis at 48 h	
Morphological criteria of entosis	CIC appearance, crescent nucleus shape of the outer cell, entotic vacuole, no membrane fusion.	
CDC42 status	Deletion of CDC42	

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
