# Peer review of "Entosis: From Cell Biology to Clinical Cancer Pathology"

_cancers, 2020, doi:10.3390/cancers12092481_

Round 1

Reviewer 1 Report

This review article addresses an interesting phenomenon and relatively novel topic of cancer biology: entosis. In principle, the article is well written. A somewhat unusual feature is the addition of experimental data (Figure 5, Tables 1 and 2) to a review article. This provides some novelty to the article which would otherwise add relatively little to recently published review articles, as outlined below. However, this presentation also provokes criticism. In my opinion, it is preferable to separate a review of the literature from the description of experimental data.

1) It is unclear to me which novel aspects of entosis would be discussed that have not been previously reviewed by Fais and Overholtzer (Nature Reviews Cancer 2018) or Durgan and Florey (BBA 2018). The authors should point out specifically in the Introduction section what their article adds to recently published excellent reviews.    

2) The experimental part mentioned above lacks information about exerimental details are provided (cell culture conditions etc.).   

Reviewer 2 Report

Comments and suggestions for authors:

The manuscript cancers-863977, by Izabela Mlynarczuk-Bialy and co-workers aims to revise the phenomenon of entosis exploring concepts of cell biology and possible importance in cancer pathology.   The authors describe general concepts relative to entosis in an attempt of revising the original concept of cell death mechanism proposing a distinct concept in cancer histopathology. After careful reading, I was no convinced of such distinction from the previous concept for the phenomenon. Also, the structure, the writing and the new data added relative to a similar review published in 2018 in the theme from J. Durgan and O. Florey lead me to consider that this revision on the theme is far from adding novelty and accurate observations to be considered.

As major comments, it is confusing and perhaps in the opposite description the explanation of the nomenclature adopted in the beginning of the introduction; where it seems that the internalizing “outer” cell should be referred as entotic whereas the internalized “inner” cell should be named as host cells. I would suggest revising thee adopted nomenclature according with the rest of the document. Also, section 3.2 is also describing cellular models as in section 3.4. I don´t understand the need to subsection 3.2 and 3.4 and the fact that section 3.3 does not exist. The type of writing, and the precision of these kind of details are important in a review on a theme and I believe that in this case, should and could be better improved.

Then, in particular;

1- the phrase starting in line 57, where is mentioned that entosis may become a novel prognosis predictor factor in cancer histopatology; this same concept is better explained in line 61. As such, the concept in line 57 can be erased, reinforcing only the same in line 61.

2- In line 67, I suggest to introduce the phrase written in line 83 to start introducing the figure before its description. I believe it is more useful to the reader.

3-In figure 2, Rock1 and 2 should be instead of Rock 1 i 2.

4-In line 117 the sentence It is known that expression of cadherin proteins E-cadherin or  P-cadherin induces entosis in human cancers should be written, for instance, isolating E-cad and P-cad from the sentence between comas.

5-In line 135 careful should be taken in stating entosis as a new predictor factor in cancer prognosis and suggesting instead that could be a new predictor factor.

6-From line 142 on the size of the letter in the document is different from the previously used.

7- I consider the information supplied in the two first paragraphs of 3.2 section redundant and not adding the necessary and commented precision and accuracy on the theme.

8- In figure 5 it is assumed by the staining of CFDA and DAPI the phenotypes described as lysosomal degradation, apoptosis without the usage of specific markers that corroborate them. Additionally, it is not described the meaning of CFDA staining. The reader does not have to know the meaning of the nomenclature. Still in this legend figure 5, it is useless and not scientific information the comparison with matrioshka.

9- the concluding remarks are somehow a repetition of previous sentences.

10- the authors claim in the beginning of 3.4 section, that originally entosis has been described in MCF7 cells in non-adherent conditions from 2007, a paper from M. Overholtzer and co-workerss.. However, this paper refers this originally described phenomenon as occurring in MCF10A, a cell line of the breast epithelium non-cancerous. This way, the reference of the authors is not correct.

In summary, based on all the concerns described above, this reviewer considers that in its present stage the manuscript is too far from providing significant advance of our understanding of the phenomenon described, does not supply novel and clarifying data and therefore it needs to be improved.

Round 2

Reviewer 1 Report

The authors have responded well to the reviewer criticism and have made appropriate changes to the manuscript. Howver, a few typos need to be corrected (e.g. "entotosis" in the legend to Figure 1, or B-catenin in Figure 2 - should be beta-catenin).

Reviewer 2 Report

New Comment on the manuscript cancers-863977, by Izabela Mlynarczuk-Bialy and co-workers:

Although changes and improvements were made in this new version of the revision on entosis, I still have detailed questions to make and since it is a revision on the theme, I consider important to be clarified for the improvement of the work. This way, I would like to clarify the following details of the manuscript:

  • In the abstract line 18; ….”proven for: head and neck squamous cell carcinoma, anal carcinoma, lung adenocarcinoma, pancreatic ductal carcinoma, as well as breast cancer…”

Still in the astract, please consider alter the word proven to shown or demonstrated; also, if it is specified for other types of carcinomas, lung, pancreatic ductal, why in the case of breast cancer is not so well specified; for example; breast carcinoma, ductal or basal?

  • Line 53- unnecessary statement; “Thus, it is not simply “devoured” by its neighbour.” Consider removal.
  • Line 63-“ One of the biggest studies was published this year and two others within last two years”; redundant phrase, consider removal.
  • Line 73- plural- Rho gtpases.
  • Line 74- known molecules and not molecular pathways as described in the legend of figure 2.
  • Line 76- what is the meaning of CIC?; specify the nomenclature the first time is used.
  • Better explanation of the symbols used in figure 2. Actin; microtubule, as in the description.

section 2.1 – what CIC structures means again?

Line 117- even escape from them? Escape from what? To elusive this sentence. Please consider revise it.

line 137 -Subsequently the internal cell can leave the outer cell intact after a long time, and the outer cell also remains intact [2]. Thus, both cells can survive enosis, leading to failure in the therapy and recurrence of cancer. –reformulate the sentence is too confuse.

In section 3.1 –table 1 conclusions confuse the description and not accurate. Please consider revise this part of table 1 and removal of table title of “in order of publication”. This does not add information since is described in table.

line 144 – number of entoses? Number of entotic particles or CTC structures. The use of these distinct names is not clear in this region of the text.

line 149- In another recent study it was also shown that entotic figures represent 91% of CIC structures in early breast cancer [29]. What is early breast cancer? not invasive? Please clarify.

line 151- However, the prognosis differs depending on the type of CIC structures and cancer. Does this mean different subtypes of breast cancer? Not matching with table 2 statement. Please clarify!

line 156 –this study is a case study case; one patient; relevance of the sentence in line 156.

Table 1 could be included before 158 line; better placed in the review.

line 166- Please verify this statement:

Those data provide an argument that increased entosis in this type of cancer indicates worse prognosis. However, the authors did not calculate the exact entosis ratio in specimens from these  EOC 268 cases.

line 170- revise spelling  with tumour progression? The answer lies in the hypothesis that entosis has a dual role.

In  Figure 3 legend please add the scale bar and what selection corresponds to the magnifications? The title of the figure should be changed to: Photograph of pancreatic cancer cells BxPC3 cultured on coverslips stained with H&E (Haematoxylin/Eosin).

 line 203- “Cell-in-Cell-in-Cell structure”- confuse!! Consider revision.

line 207- In cancer cell lines in culture instead of In cell culture of cancer cells.

Figure 4- not explained what are the magnifications of the figure and which region it represents; should be corrected, scale bars should be added and the staining is Hematoxilin & eosin (H&E).

Figure 5-  Entosis in BxPC3 cells cultured on glass. Reformulate Entosis in BxPC3 cells. Why it needs to add cultured on glass?

line 226: In addition, none of the cells shows degenerative features such as cell nuclei fragmentation, chromatin condensation, or lack of continuity of cell membranes. Misplaced the underlined word.

line 230 – scale  bar 10µM. Is 10 mm and not mM that means millimolar and not the unit micra.

line 241_ correct as: Overholtzer for the MCF7 cells using the pancreatic cancer cell  line BxPC3.

line 251 - and. regressive  changes (i.e. degeneration, both nucleus and/or cytoplasm alternations), cell division. are Detailed morphology of entotic cells are shown in Figure 5.- reformulate. This is very confuse, with type errors in the text. Please revise.

line 258- In BxPc3  -cell model 80 % of inner entotic cells 80% of the 3% entotic figures in BxPc3? Is this correct? The impact of 80% in 3% entotic figures is very different from the 70% of 30% in MCF7! Consider revise the sentence concerning the impact of the comparison.
